# Metabolic Profile, Biotransformation, Docking Studies and Molecular Dynamics Simulations of Bioactive Compounds Secreted by CG3 Strain

**DOI:** 10.3390/antibiotics11050657

**Published:** 2022-05-13

**Authors:** Omar Messaoudi, Enge Sudarman, Chirag Patel, Mourad Bendahou, Joachim Wink

**Affiliations:** 1Department of Biology, Faculty of Science, University of Amar Telidji, Laghouat 03000, Algeria; o.messaoudi@lagh-univ.dz; 2Microbial Strain Collection, Helmholtz Centre for Infection Research GmbH (HZI), Inhoffenstrasse 7, 38124 Braunschweig, Germany; 3Laboratory of Applied Microbiology in Food and Environment, Abou Bekr Belkaïd University, Tlemcen 13000, Algeria; bendahou63@yahoo.fr; 4Department Microbial Drugs, Helmholtz Centre for Infection Research GmbH (HZI), Inhoffenstrasse 7, 38124 Braunschweig, Germany; e.sudarman@web.de; 5German Centre for Infection Research Association (DZIF), Partner Site Hannover-Braunschweig, Inhoffenstrasse 7, 38124 Braunschweig, Germany; 6Computer-Aided Drug Design Group, Chemical Biology Laboratory, Center for Cancer Research, National Cancer Institute, National Institute of Health, Frederick, MD 21702, USA; chiragpatel269@gmail.com

**Keywords:** *Nocardiopsis* CG3, bioactive compound, biotransformation, isoflavones, aromatase, molecular docking, dynamics simulations, breast cancer

## Abstract

*Actinobacteria* isolated from untapped environments and exposed to extreme conditions such as saltpans are a promising source of novel bioactive compounds. These microorganisms can provide new molecules through either the biosynthetic pathway or the biotransformation of organic molecules. In the present study, we performed a chemical metabolic screening of secondary metabolites secreted by the new strain CG3, which was isolated from a saltpan located in the Sahara of Algeria, via high-performance liquid chromatography coupled with high-resolution mass spectrometry (HPLC-ESI-HRMS). The results indicated that this strain produced five new polyene macrolactams, kenalactams A–E, along with two known compounds, mitomycin C and 6″-hydroxy-4,2′,3′,4″ tetramethoxy-p-terphenyl. Furthermore, the CG3 isolate could have excellent properties for converting the aglycone isoflavone glycitein to the compounds 6,7-dimethoxy-3-(4-methoxyphenyl)chromen-4-one (**50**) and 6,7-dimethoxy-3-phenylchromen-4-one (**54**), and the isoflavone genistein can be converted to 5,7-dimethoxy-3-(4-methoxyphenyl)chromen-4-one (**52**). Docking studies and molecular dynamics simulations indicated that these three isoflavones, generated via biotransformation, are potent inhibitors of the target protein aromatase (CYP19A1); consequently, they can be used to prevent breast cancer risk in postmenopausal women.

## 1. Introduction

In recent years, research on novel bioactive compounds has been more oriented towards the exploration of secondary metabolites secreted by *Actinobacteria* isolated from untapped environments exposed to extreme conditions [1]. Such environments offer the capacity to isolate diverse novel microorganisms, which are supposed to be potential reservoirs of novel compounds. Saltpans located in the Sahara of Algeria are one of these extreme environments, which are characterized by high temperature in the morning, severe solar radiation, low temperature at night, very low amounts of nutrients and high salt concentration [2]. *Actinobacteria* has adapted to such extreme conditions, acquiring complex hypha by which bioactive compounds can be produced [3]. Little is known about the diversity of actinomycetes isolated from saltpans located in the Sahara of Algeria [2,4,5]; for this reason, we focus our search on these extreme environments in order to increase the discovery rate of new *Actinobacterial* strains that can provide new metabolites.

*Actinobacteria* are the most potent source of bioactive secondary metabolites, which display different activities. These compounds can be obtained in two ways: (i) The first way is the biosynthetic pathway: in this case, *Actinobacteria* use different enzymes, such as polyketide synthase (PKSI, PKII and PKSIII) [6], non-ribosomal peptide synthase (NRPS) [7] and the hybrid PKS-NRPS [8], for the biosynthesis of secondary metabolites. PKS enzymes use an extender unit, which is usually an acyl-COA such as malonyl-CoA, methylmalonyl-CoA, ethylmaloyl-CoA or methoxymalonyl-COA, for the biosynthesis of polyketide molecules [9]. However, NRPS enzymes use different monomers of proteinogenic and nonproteinogenic amino acids for the biosynthesis of non-ribosomal peptides [10]. Furthermore, hybrid enzymes (PKS–NRPS) produce polyketide–amino acid compounds with striking structural features. (ii) The second way is biotransformation, which allows modifying the structure of organic molecules with specific enzymes, enabling the catalysis of different reactions [11], including hydroxylation, dehydroxylation, methylation and demethoxylation. This method is a great alternative to organic synthesis and offers the advantage of high stereoselectivity [12].

During our investigation of new taxa of actinomycetes producing new bioactive metabolites, five new compounds, kenalactams A–E, were isolated from the new strain CG3, which was isolated from a saltpan located in the Sahara of Algeria [13]. In order to continue the exploration of secondary metabolites secreted by strain CG3, chemical metabolic screening was applied on the basis of high-performance liquid chromatography coupled with high-resolution mass spectrometry (HPLC-ESI-HRMS).

Herein, we report: (i) the bioactivity evaluation of CG3 crude extracts in two different mediums, ISP2 and soybean medium (SM); (ii) the bioassay-guided fractionation of the crude extract from SM via analytical HPLC against a panel of human pathogenic microorganisms; (iii) metabolic studies for the identification of bioactive secondary metabolites secreted by strain CG3 using HPLC-ESI-HRMS; and (iv) an in silico study of three isoflavone derivatives obtained via biotransformation as inhibitors of the target protein aromatase.

## 2. Results and Discussion

### 2.1. Antimicrobial Activity of Strain CG3

The MIC values for the antimicrobial activity of strain CG3 against the tested pathogenic microorganisms are shown in Table 1.

The results revealed that extracts prepared from SM and ISP2 media exhibited strong antibacterial activity against all tested bacteria, except for *Klebsiella pneumoniae*, with MIC values ranging between 0.52–66.66 µg/mL. However, only the extract obtained from SM showed the inhibition of *Candida albicans* and *Mucor hiemalis*, with MIC values of 66.66 µg/mL and 16.66 µg/mL, respectively. The difference in activity between ISP2 and SM can be attributed to the difference in the composition of the two culture media. This composition, particularly the nature of carbon and nitrogen sources, can affect the quality and the quantity of bioactive compounds secreted by strain CG3. Therefore, SM met the nutritional requirements of strain CG3 for the biosynthesis of bioactive molecules with antifungal and antibacterial activities, whereas only metabolites with antibacterial activity were secreted by strain CG3 when it was grown in ISP2 medium.

### 2.2. Fractionation of Crude Extract

In order to localize the inhibitory activity, the crude extract from SM was fractionated via analytical HPLC and tested against *Staphylococcus aureus* (Figure 1A) and *Mucor hiemalis* (Figure 1B) and subsequently analyzed by HPLC-ESI-HRMS.

The results in Figure 1 indicate the presence of three active areas, (1D, 1F), (3F, 3H) and (4B, 4D), in 96-well microplates. By correlation with the analytical HPLC chromatogram, the active fractions corresponding to the peaks were located at the intervals of retention times: 2–2.40 min, 10.50–11.30 min and 14.50–16.50 min, respectively. Furthermore, the two active peak areas eluted at *t*_R_: 2–2.40 min and *t*_R_: 10.50–11.30 min were active against *Staphylococcus aureus*, whereas the third one at *t*_R_: 14.50–16.50 min exhibited antifungal activity against *Mucor hiemalis*.

The comparison between the analytical HPLC chromatographic profiles of the crude extracts from SM (Figure 1) and ISP2 media (Appendix A) indicated that the three peaks eluted between *t*_R_: 14.30–16.50 min were absent in the crude extract from ISP2 medium (Appendix A). These peaks are responsible for the antifungal activity of the crude extract prepared from SM against *Mucor hiemalis* (Figure 1).

### 2.3. HPLC-UV-HRESIMS Analysis of Crude Extracts

In order to identify the interesting peaks, the crude extract prepared from SM was analyzed by HPLC-UV-HRESIMS (Figure 2).

In order to correlate the peaks from Figure 2 with those in Figure 1, we follow two steps: (i): The first step consists of comparing the chromatographic profiles in both Figure 1 and Figure 2, which are shown to be similar (almost the same morphology). Therefore, each peak in Figure 2 can be correlated to its homologue in Figure 1 according to its position in the chromatogram. (ii): To confirm the correlations, in the second step, comparison of the UV-Vis spectrum of each peak in Figure 1 (measured by analytical HPLC) to the UV-Vis spectrum of the corresponding peak in Figure 2 (measured by HPLC-UV-HRESIMS) is carried out. The UV profile of the peaks in Figure 1 should be exactly identical/similar to that of the corresponding peaks in Figure 2.

By comparing the HPLC chromatographic profile in Figure 1 with the LC-UV-MS chromatographic profile in Figure 2, the first zone (1D, 1F) of activity in Figure 1 (*t*_R_ = 2–2.40 min) was assigned to peak numbers **12** (*t*_R_ = 3.77 min) and **13** (*t*_R_ = 3.95 min) in Figure 2. Peaks **12** and **13** exhibited the same UV-Vis absorption maxima at 216, 245 and 362 nm and the same molecular ion cluster [M+H]^+^ at *m*/*z* 368.2390, which provided the same molecular formula of C_23_H_29_NO_3_. This indicates that both metabolites (**12** and **13**) are isomers.

The search for molecules with the same character in natural chemical product databases, such as Antibase and Dictionary of Natural Products, revealed that compounds **12** and **13** are correlated to mitomycin C and its stereoisomers (Figure 3).

Mitomycin C is a methylazirinopyrroloindoledione purified from *Streptomyces caespitosus* and presents potent antineoplastic activity against a variety of cancers due to its ability to crosslink DNA with high efficiency [13]. Furthermore, this compound has a wide spectrum of antibacterial activities [14]; therefore, the antimicrobial activity against *Staphylococcus aureus* observed in Figure 1 at *t*_R_: 2.00–2.40 min can be correlated to mitomycin C. Mitomycin C has been previously isolated from several strains belonging to *Streptomyces*. However, it should be noted that this compound was identified for the first time in the crude extract prepared from the culture fermentation of a strain closely related to the genus *Nocardiopsis*.

Peak numbers **39** (*t*_R_ = 8.80 min) and **43** (*t**_R_* = 7.79 min) (Figure 2) correspond to the active area of (3F, 3H) in Figure 1. Both compounds exhibited the same UV-Vis absorption maxima at 248, 297 and 339 nm and the same molecular ion cluster [M+H]^+^ at *m*/*z* 368.2390. The molecular formula of both compounds was determined to be C_23_H_29_NO_3_ by HPLC-UV-HRESIMS.

The structures of **39** and **43**, assigned from 1D and 2D NMR spectra (^1^H ^1^H COSY, ^1^H ^13^C HMBC and HSQC), revealed that both molecules form 22-membered macrocyclic amide rings; therefore, they belong to the class of polyene macrolactams (Figure 3) [15]. The names kenalactams A and B are attributed to **43** and **39**, respectively (Figure 3). Due to the low amount and instability of compound **39,** only compound **43** was tested for antimicrobial and cytotoxic activities at concentrations ≥ 66.66 μg/mL, and the results indicate that **43** did not show any antimicrobial activity against *Staphylococcus aureus* [15]. Consequently, the antibacterial activity in Figure 1 at *t*_R_: 10.50–11.30 min can be linked to kenalactam B (**39**).

The third active area (4B, 4D) eluted between *t*_R_: 14.50–16.50 min and exhibiting antifungal activity against *Mucor hiemalis* (Figure 1) corresponds to peaks **50**, **52** and **54** in Figure 2. These three peaks showed the same UV-Vis absorption maxima at 220, 254 and 310 nm.

Compound **50** was isolated from the crude extract of strain CG3 in SM. HPLC-UV-HRESIMS analysis of **50** revealed a molecular ion cluster [M+H]^+^ at *m*/*z* 313.1074 with the molecular formula C_18_H_16_O_5_, implying the presence of 11 unsaturation. Its complete structure was determined by 1D and 2D NMR spectroscopy. The ^13^C NMR and ^1^H ^13^C HSQC NMR spectral data (Appendix A) confirmed the presence of 18 carbons, including 7 olefinic methines (δ_H_ 7.01–8.24, δ_C_ 154.9–101.2), 7 quaternary olefinic carbons (δ_C_ 177.9–118.7), including 1 carbonyl carbon at δ_C_ 177.9, and 3 methoxy carbons (δ_H/C_ 3.86/55.9, 3.96/56.8 and 4.00/57.2).

Further analysis of 2D NMR spectra (HSQC-DEPT, COSY and HMBC) revealed the presence of 6,7-dimethoxy-3-(4-methoxyphenyl)chromen-4-one. This was confirmed by comparing the NMR spectroscopic data with those reported in the literature [16].

The molecular formula of **54** was determined by HPLC-UV-HRESIMS to be C_17_H_14_O_4_ with a molecular ion cluster [M+H]^+^ at *m*/*z* 283.0967. The ^1^H NMR and ^13^C NMR data of compound **54** (Appendix A) contained all structure elements of **50**, and the only difference was the absence of methoxy carbons (δ_H/C_ 3.86/55.9) linked to the C-4′ position; therefore, compound **54** was identified as 6,7-dimethoxy-3-phenylchromen-4-one. However, the molecular formula of C_18_H_16_O_5_ of compound **52** with a molecular ion cluster [2M+Na]^+^ at *m*/*z* 647.1900 was determined by HPLC-UV-HRESIMS. A survey of chemical databases (Antibase and Dictionary of Natural Products) of compounds with similar properties identified compound **52** as 5,7-dimethoxy-3-(4-methoxyphenyl)chromen-4-one.

To the best of our knowledge, compounds **50**, **52** and **54** have never been tested to evaluate their antifungal activity, and the antifungal activity seen in Figure 1 at *t*_R_: 14.50–16.50 min against *Mucor hiemalis*, as well as the antifungal activity of the crude extract prepared from the culture of strain CG3 in SM (Table 1), can be attributed to one of these three compounds or to the synergistic action between them.

Compounds **50**, **52** and **54**, belonging to the isoflavone subclass, are isomers of flavones. Both subclasses, isoflavones and flavones, pertain to the class of flavonoids, and they share an almost identical structure. The only difference in the structure compared to flavones is the position of the phenyl group, which is linked to C-3 instead of C-2 for flavones [17,18]. Flavonoids are polyphenolic plant-derived secondary metabolites, generally found in various plant species [18].

Isoflavones **50**, **52** and **54** were detected only in the crude extract prepared from the culture of strain CG3 in SM (Appendix A). On the other hand, the three compounds were completely absent when the crude extract from ISP2 medium was analyzed by HPLC-UV-HRESIMS (Appendix A). SM contains soybeans (20.0 g/L); however, this ingredient is absent in ISP2 medium. The soybeans in SM are the major natural source of isoflavones [19]; therefore, compounds **50**, **52** and **54**, detected in the crude extract of SM, likely originate from the culture medium and are not biosynthesized by strain CG3. To exclude this hypothesis, SM was incubated without strain CG3 at 37 °C for 14 days. The HPLC-UV-HRESIMS analysis of the crude extract prepared from this last culture indicated the absence of compounds similar to metabolites **50**, **52** and **54** in SM (Appendix A).

Twelve isoflavones, including three aglycone isoflavones (daidzein, genistein and glycitein) and nine glycosylated isoflavones possessing a sugar moiety at C-7, were identified in soybean [17].

Compounds **50**, **52** and **54** are aglycone isoflavones. Glycitein is the structure most closely related to **50** and **54**; in fact, the three compounds have a methoxy group linked to the C-6 position of the isoflavone core. However, the two hydroxy groups of glycitein attached to C-7 and C-4′ are methylated in **50**. Compound **54** is methylated at C-7, in addition to C-6, whereas the methoxy group of **50** linked to C-4′ is absent in **54** (Figure 3).

Compound **52** is structurally close to genistein; the only difference is that the hydroxyl groups at C-5, C7 and C-4′ in genistein are replaced by methyl groups in **52** (Figure 4).

We can expect that strain CG3 generates compounds **50**, **52** and **54** via biotransformation of the aglycone isoflavone, genistein and glycitein (Figure 4), which were detected in the crude extract prepared from SM without the CG3 strain (*t*_R_ = 4.40–7.50 min) (Appendix A). In fact, we suppose that strain CG3 generates compound **50** in the first step via the O-methylation of both hydroxyl groups of glycitein, attached at positions 7 and 4′ (Figure 4). These reactions would be catalyzed by methyl transferase enzymes such as 7-O-methyltransferase (7-OMT) and 4′-O-methyltransferase (4′-OMT), respectively. The source of methyl groups linked to the 4′-O and 7-O positions is S-Adenosyl-L-methionine (SAM), which is one of the major methyl donors in all living organisms [20] (Figure 5).

Compound **54** was possibly generated by strain CG3 through the direct demethoxylation of **50** at the 4′-O position (Figure 4). In fact, after a long period of incubation (up to 21 days), the amount of **50** decreased gradually, whereas the production of metabolite **54** increased *(*Appendix A).

Setchell et al. [21] explained that the isoflavone daidzein can be generated via demethoxylation of glycitein at the 6-position. This reaction (demethoxylation) is a minor biotransformation pathway.

Furthermore, methylation of genistein at C-6, C-7 and C-4′ led to the formation of compound **52** (Figure 4).

Compounds **52** and **54** were obtained via organic synthesis using eleven phenol derivatives and six phenylacetic acids [22]. However, compound **50** was generated from the chemical transformation of the isoflavone 6-methoxyisoformononetin, which was isolated from the roots of the medicinal plant *Amphimas pterocarpoides* [16].

Isoflavones have been considered chemoprotective compounds; consequently, they can be used to reduce the risk of a wide range of chronic diseases, such as diabetes, osteoporosis and cardiovascular diseases. Additionally, they may protect the body from hormone-related cancers, such as breast cancer and prostate cancer. Seo et al. [23] indicated that compound **50** acted synergistically with glycitin to promote wound healing and reduce scarring. Therefore, it could potentially be developed in conjunction with glycitin as a bioactive therapeutic agent for wound treatment.

It should be noted that compounds **50**, **52** and **54**, which have previously been reported as synthetic compounds, are described for the first time as natural products from the fermentation of CG3.

Metabolite **65**, appearing at the retention time of 12.00 min, exhibited UV-Vis absorption maxima at 222 nm and 268 nm, and its molecular formula was determined to be C_22_H_22_O_5_ on the basis of its molecular ion clusters [M+H]^+^ at *m*/*z* 367.1546 and [2M+Na]^+^ at *m*/*z* 7,552,834.

Compound **65** showed ^1^H-NMR and ^13^C-NMR data identical to those of the known compound 6′-hydroxy-4,2′,3′,4′-tetramethoxy-*p*-terphenyl reported in the literature [24] (Appendix A), indicating the existence of a tetramethoxy- *p*-terphenyl analog. Detailed analysis of 1D and 2D NMR data confirmed the structure of 6′-Hydroxy-4,2′,3′,4″-tetramethoxy-*p*-terphenyl.

The *p*-terphenyl derivative **65** was isolated for the first time from the halophilic actinomycete *Nocardiopsis gilva* YIM 90087, which is closely related to strain CG3 [24].

Compound **69** (*t*_R_ = 12.43 min) was found to have a molecular ion cluster [M+H]^+^ at *m*/*z* 459.3012 with the molecular formula C_30_H_38_N_2_O_2_, whereas compounds **75** (*t*_R_ = 13.25 min) and **78** (*t*_R_ = 14 min) exhibited a molecular ion cluster [M+H]^+^ at *m*/*z* 499.3323 with the molecular formula C_33_H_42_N_2_O_2_ and [M +H]^+^ at *m*/*z* 539.3636 with C_36_H_46_N_2_O_2_, respectively. Structure elucidation based on the NMR and HPLC-UV-HRESIMS data of these three metabolites isolated from the CG3 strain was described previously by Messaoudi et al. [15], and these unprecedented compounds, named kenalactams C–E (**69**, **75** and **78**), were identified as the first family of polyenic macrolactams from the genus of *Norcadiopsis*.

Compounds **69**, **75** and **78** exhibited weak to moderate antimicrobial and cytotoxic activities against a panel of human pathogenic microorganisms and human cancer cell lines. Compound **78** was the most potent, showing inhibition against KB3.1, PC-3, SKOV-3 and A549 cell lines in an IC_50_ range of 2−5.5 μM [15].

### 2.4. Molecular Docking Using AutoDock

Three isoflavone derivatives, **50**, **52** and **54**, are structurally related to estrogen steroid sex hormones, such as estrone (E1), estradiol (E2), estriol (E3) and estetrol (E4) [25]. Therefore, they can exhibit estrogenic and/or antiestrogenic effects [26].

Estrogenic hormones in females are synthesized primarily by the ovaries from androgens, such as testosterone and androstenedione. The reactions are catalyzed by the cytochrome P450 19A1 (CYP19A1; EC 1.14.14.1), commonly known as aromatase. Estrogens are also produced in smaller amounts by other tissues, such as breasts [27].

The quantity of estrogens produced in female organisms is related to the emergence of some kinds of cancers, particularly estrogen-dependent breast cancer. In this case, the attachment of estrogen to specific estrogen receptors releases a signal, which stimulates the proliferation of breast cancer tumor cells. Approximately 80% of breast cancer is estrogen receptor (ER)-positive [28].

Blocking the enzyme aromatase through the use of aromatase inhibitors is able to stop the production of estrogen, which contributes to decreased breast cancer risk [29]. One group of potent inhibitor compounds is isoflavones, which can be used to decrease cancer risk by inhibiting aromatase enzyme activity and CYP19 gene expression in human tissues [30].

It should be noted that aromatase inhibitors cannot stop estrogen production in the ovaries, but they block their synthesis in secondary sources of estrogens, particularly in the breast; consequently, aromatase inhibitors are only effective for postmenopausal women [31].

Molecular docking using AutoDock Tools 1.5.4 was performed in order to evaluate the inhibitory potential of **50**, **52** and **54** against the target protein aromatase (CYP19A1). Obtained binding modes and docking energies of **50**, **52** and **54** were compared with those of the reference androgen, androstenedione. The results of the free binding energies (ΔGb), calculated by AutoDock, are summarized in Table 2.

Based on the AutoDock results (Table 2), **50**, **52** and **54** exhibited docking scores of −7.1 kcal/mol, −7.5 kcal/mol and −7.3 kcal/mol, respectively. However, the reference, androstenedione, displayed a binding energy of −9.8 kcal/mol with the active site of the target protein aromatase.

The crucial amino acids located at the active site of the enzyme aromatase were identified as Ile305, Ala306, Ala307, Asp309, Thr310 Arg115, Ile133, Phe134, Phe221, Trp224, Val369, Val370, Leu372, Val373, Met374, Leu477 and Ser478 [32,33]. Furthermore, these residues were found to occupy an inner cavity volume of 1525.92 Å^3^ with an entrance of 3.24 Å in diameter. Hence, only small molecules with low molecular weight can penetrate the binding cleft [34].

A careful analysis of the enzyme binding pocket indicated that it is highly hydrophobic, and it was assembled by the condensation of non-polar aliphatic amino acid residues. Therefore, only inhibitors bearing alkyl or aromatic groups are expected to bind with high affinity [35].

Compounds **50**, **52** and **54** are small hydrophobic molecules consisting of two benzene rings linked by a heterocyclic pyran ring. These interact with the protein CYP19A1, mainly through hydrophobic and some hydrogen bonds (Figure 6).

The most potent ligand, **52**, was found to exhibit hydrophobic alkyl interactions with the key residues Leu477, Met374, Val370, Ala306 and Ile133 at the active site, whereas a H-bond interaction was observed between the NH group of Arg115 and the oxygen-bearing ketone group of **52**, which tend to form a stable binding interaction (Figure 6A). Compounds **50** and **54** interact with important amino acids in the active site of the target CYP19A1, specifically via electrostatic and hydrophobic interactions. H-bonding to the target CYP19A1 was not observed for either **50** or **54** (Figure 6B,C). In addition, the reference hormone, androstenedione, which fits within the active site cavity, displayed two hydrogen bonds with Arg115 and Met374, one unfavorable donor–donor bond with Asp309 and one carbon–hydrogen bond with Ala306. However, the two residues Trp224 and Val370 were involved in the alkyl interaction (Figure 6D).

### 2.5. Molecular Dynamics Simulation

Molecular dynamics (MD) simulations were performed for a 100 ns time interval to understand the dynamic behavior and evaluate the stability of the docked complexes 3EQM–**50**, 3EQM–**52** and 3EQM–**54**. Following the completion of MD simulations, a root mean square deviation (RMSD) evaluation was carried out and used to measure the natural change in particle removal for a given frame as compared to a standard constant frame. Protein–ligand RMSD values are depicted in Figure 7.

The RMSD plots in Figure 7 reveal that the complex 3EQM–**50** (Figure 7A) stabilized after 35 ns, while the complex 3EQM–**54** (Figure 7C) remained stable over the trajectory of the 100 ns simulation, with a slight perturbance near 78 ns. However, the complex 3EQM–**52** (Figure 7B) showed stability from initiation to the end of the MD simulation event. Furthermore, the RMSD values of the three complexes remained under the 3 Å range, which is within the acceptable region and indicates that these complexes were stable throughout the simulation.

Figure 8 illustrates the ligand properties, including RMSD, rGyr (radius of gyration), intraHB (intramolecular hydrogen bonds), MolSA (molecular surface area), SASA (solvent-accessible surface area) and PSA (polar surface area), of all MD complexes.

The RMSD values for the three complexes 3EQM–**50** (Figure 8A), 3EQM–**52** (Figure 8B) and 3EQM–**54** (Figure 8C) remained under 1 Å with little deviation. This indirectly indicates fewer conformational fluctuations and greater stability of the three complexes within the pockets of the human placental aromatase cytochrome P450 target during the MD study. The stabilization of the inhibitors **50**, **52** and **54** was supported by rGyr (4.22–4.35, 4.08–4.23 and 3.74–3.86) Å, PSA (70.02–85.02, 67.0–78.0 and 59.0–73.0) Å^2^, SASA (5.0- 165.0, 2.0–37.5 and 1.0–40.0) Å^2^ and MolSA (297.5–305.3, 297.5–305.3, 271.0 and 278.0) Å^2^, respectively (Figure 8).

Intramolecular interactions of **50**, **52** and **54** with the human placental aromatase cytochrome P450 target are observed in Figure 9.

Hydrophobic interactions were identified with higher numbers during the MD simulation event for the three inhibitors (Figure 9). Furthermore, the interactions of the residues of the target protein, 3EQM, with the three inhibitors **50**, **52** and **54** showed that Phe134 and Met374 interacted with **52** through a hydrophobic moiety bond and hydrogen bond through a water molecule for 58% and 45% of the simulation time, respectively (Figure 9B). However, the catalytic residue Phe134 of 3EQM formed a hydrophobic interaction with **54** for 46% of the simulation time (Figure 9C).

## 3. Materials and Methods

### 3.1. Selective Isolation of CG3 and Identification

Strain CG3 was isolated from soil samples collected at a depth of 15–20 cm below the surface of saltpan located in Kenadsa region (31°31′59.99″ N, 2°24′59.99″ E), Bechar province (Southwestern of Algeria), using Starch Casein Agar (SCA) medium, after incubation at 37 °C for 14 days [36].

Molecular identification of strain CG3, based on the sequencing of 16s rDNA, indicated that this isolate is closely related to *Nocardiopsis rosea* YIM 90094^T^ (99.2%). Strain CG3 was distinguished from its closest species based on the results of phylogenetical, morphological, physiological and biochemical characterization. Therefore, strain CG3 was identified as a novel species within the genus *Nocardiopsis* [36].

### 3.2. Evaluation of Antimicrobial Activity

#### 3.2.1. Preparation of Suspension

Antimicrobial activity was evaluated against ten different microorganisms, including eight bacteria and two fungi, which displayed a wide array of differences in their taxonomic positions, morphological and physiological characteristics, and their virulence.

Among bacteria, four are Gram-positive and belong to two different phyla, among which three are *Firmicutes* (*Micorococcus luteus* DSM1790, *Staphylococcus aureus* Newman and *Bacillus subtilis* DSM10), and one belongs to the *Actinobacteria* phylum (*Mycobacterium smegmatis* ATCC 700084). This last one is a fast grower and non-pathogenic model for research on a new anti-tuberculosis drug. In addition, four Gram-negative bacteria were used, which all belong to the phylum of *Proteobacteria (Chromobacterium violaceum* DSM 30191, *Pseudomonas aeruginosa* PA14, *Klebsiella pneumoniae* ATCC and *Escherichia coli* TolC).

Three of the bacteria used belong to the ESKAPE group, namely, *Pseudomonas aeruginosa* PA14, *Klebsiella pneumoniae* ATCC and *Staphylococcus aureus* Newman, which includes highly virulent and antibiotic-resistant pathogens. Furthermore, two fungi were used, including one yeast (*Candida albicans* DSM1665) and one mold (*Mucor hiemalis* DSM 2656).

The selected microorganisms were obtained from DSMZ (German Collection of Microorganisms and Cell Cultures), Braunschweig, Germany, and ATCC (American Type Culture Collection), Manassas, VA, USA.

Suspension of the pathogenic microorganisms was prepared by inoculation of bacteria in EBS medium (0.5% casein peptone, 0.5% glucose, 0.1% meat extract, 0.1% yeast extract, 50 mM HEPES [11.9 g/L], pH 7.0), while fungi were inoculated in MYG medium (1.0% phytone peptone, 1.0% glucose, 50 mM HEPES [11.9 g/L], pH 7.0). After 24 h incubation at 37 °C or 30 °C for bacteria and 48 h incubation at 30 °C for fungi, the turbidity of each test microorganisms was adjusted to 0.05 McFarland for bacteria and 0.01 McFarland for fungi, respectively [37].

#### 3.2.2. Preparation of Crude Extract

Two Erlenmeyer flasks (250 mL), each containing 100 mL of soybean medium (SM) (2% soybean, 2% mannitol, 0.4% glucose, 3% NaCl, pH 7) and 100 mL of ISP2 medium (glucose 4.0 g/L, yeast extract 4.0 g/L, malt extract 10 g/L, distilled water 1000 mL, pH 7), were inoculated with one piece (1 cm^3^) of well-sporulated culture of CG3 strain grown on starch casein agar [38].

After two weeks of incubation in a rotary shaker at 37 °C and 160 rpm, 20 mL of each culture was taken and mixed with 20 mL of ethyl acetate in two Falcon tubes (50 mL). The two tubes were shaken for 20 min, followed by a centrifugation step at 9000 rpm for 10 min. The ethyl acetate was evaporated at 40 °C using a rotary evaporator, and the residue was dissolved in 1 mL of methanol and then centrifuged at 14,000 rpm for 10 min [39].

#### 3.2.3. Serial Dilution Method for Antimicrobial Activity

Minimum inhibitory concentrations, corresponding to the lowest concentration of the tested extract that prevents visible growth of tested microorganisms, were determined by serial dilution in 96-well microplate. Twenty-microliter aliquots with a concentration of 1 mg/mL (the final concentration in the first well was 67 μg/mL) of crude extracts prepared from the culture of strain CG3 in SM and ISP2 media were tested against different tested microorganisms. Oxytetracycline and nystatin were used as positive controls for antibacterial and antifungal activities, respectively, while methanol was used as a negative control [40].

### 3.3. Analytical HPLC and Fractionation of Crude Extract

Analytical RP-HPLC and fractionation were conducted with an Agilent 1260 HPLC system equipped with a fraction collector. Detection of peaks was performed using a diode-array UV detector (DAD-UV, Santa Clara, CA, USA) or a Corona Ultra detector (Dionex, Germering, Germany). Analytical HPLC conditions: column 100 × 2.1 mm XBridge C_18_, 3.5 μm, (Waters, Milford, CT, USA); solvent A: H_2_O−acetonitrile (95/5), 5 mmol NH_4_OAc, 0.04 mL/L CH_3_COOH; solvent B: H_2_O−acetonitrile (5/95), 5 mmol NH_4_OAc, 0.04 mL/L CH_3_COOH; gradient system: 10% B, increasing to 100% B in 30 min and maintaining 100% for 10 min; flow rate: 0.3 mL/min; 40 °C.

In order to localize the biological activity, the crude extract prepared from the culture of strain CG3 in SM was fractionated. Each fraction was collected in a 96-well microplate every 30 s. The solvent was removed with heated nitrogen in MiniVap (Porvair Sciences, Wrexham, UK) for 45–60 min at 40 °C. Afterward, each well of the 96-well microplate was inoculated with 150 µL of the suspension prepared from the former inhibited test microorganism. After incubation at 30 °C for 24 h, the inhibited wells can be correlated to the retention time (*t*_R_) and the corresponding peak.

### 3.4. Metabolic Profile

Metabolomics is an approach used to study metabolites secreted by microorganisms as well as plants; it can be defined as the “systematic study of the unique chemical fingerprints that specific cellular processes leave behind”. The metabolic profile is determined using different techniques; however, high-performance liquid chromatography coupled with high-resolution mass spectrometry (HPLC-ESI-HRMS) is a powerful analytical tool for metabolic profiling that can detect a wide range of chemical compounds at the same time without purification [41,42].

The crude extract prepared from the culture of strain CG3 in SM was analyzed by LC-HRESIMS, and results were recorded on a MaXis ESI-TOF mass spectrometer (Bruker) equipped with an Agilent 1260 series RP-HPLC system: column 50 × 2.1 mm Acquity UPLC BEH C_18_ (Waters); solvent A: 0.1% formic acid in H_2_O, B: 0.1% formic acid in acetonitrile; gradient system: 5% B for 0.5 min, increased to 100% B in 19.5 min and maintained at 100% B for 5 min; flow rate: 0.6 mL/min; 40 °C; DAD-UV detection at 200−600 nm. Molecular formulas were calculated, including the isotopic pattern, with the SmartFormula algorithm (Bruker, Billerica, MA, USA). Detected compounds were identified by comparison of molecular weight, molecular formula and UV-Visible spectrum with already-known compounds registered in chemical databases, such as Dictionary of Natural Products, which is the most comprehensive resource of natural chemical products.

### 3.5. Docking Studies and Molecular Dynamics Simulations

#### 3.5.1. Protein and Ligand Preparation

The crystal structure of the target protein, aromatase, was retrieved from Research Collaboratory for Structural Bioinformatics (RCSB) in PDB format (PDB code: 3EQM). The aromatase protein structure was prepared for docking using BIOVIA Discovery Studio Visualizer 2020 after removal of water molecules and the original ligand attached to the target protein. Polar hydrogen atoms and Kollman charges were added using AutoDock Tools 1.5.6. Finally, the target protein was saved as a PDBQT file, whereas the three ligands, **50**, **52** and **54**, were prepared for docking by energy minimization using the Gasteiger algorithm, detecting root and a set of torsions [43].

#### 3.5.2. Molecular Docking Analysis

Molecular docking calculations were performed with the program AutoDock Tools 1.5.4 using the Lamarckian Genetic Algorithm. AutoGrid was used to generate a grid box size of 50 × 64 × 78 Å points with a grid spacing of 0.375 Å, centered at x, y and z coordinates of 83.35, 49.60 and 50.60, around the hotspot residues in the active site of the target 3EQM [34].

The employed docking parameters for each docked compound were derived from 100 independent docking runs that were set to terminate after a maximum of 2.5 × 10^6^ energy evaluations with mutation rate of 0.02 and crossover rate of 0.8. The population size was set to 250 randomly placed individuals. The Lamarckian genetic algorithm was used, and the output was saved in docking parameter file (DPF) format. The predicted binding poses for each compound were processed by 0 clustering analysis (1.0 Å RMSD tolerance), and the lowest energy conformation from the largest cluster was selected as representative. Discovery Studio and PyMOL were implemented to visualize and scrutinize the interactions between the ligand fragments and aromatase protein [34].

#### 3.5.3. Molecular Dynamics Simulation

Molecular dynamics simulation was used to assess the physical motions of atoms and molecules in a protein–ligand docked complex. Desmond [44] (Schrödinger Release March 2019) was used to run molecular dynamics simulations with the human placental aromatase cytochrome P450 and three compounds. A simulation length of up to 100 ns for each run was conducted, and different parameters were computed to determine if the systems were stable. To allow complex relaxation, these complexes were prepared using a protein preparation wizard. Certain predefined parameters were considered for the simulation of cell preparation, which includes adding hydrogens, removing water, assigning bond orders, and filling in missing side chains and loops with optimization of hydrogen-bond assignment (sampling of water orientations and use of pH 7.0). The simulation periodic box was created with the System Builder module and transferable intermolecular potential with 3 points (TIP3P) water model and an all-atom force field from optimized potentials for liquid simulations (OPLS). An orthorhombic box shape with dimensions of 10∗10∗10 was used to define boundaries for the Na and Cl^-^ neutralization process. The NPT ensemble (number of atoms, pressure and temperature were constant) contains 300 K temperature and 1.01325 bar pressure to equilibrate the unrestrained processing process for 100 ns time interval. The isotropic Martyna–Tobias–Klein barostat (relaxation time = 2 ps) and the Nosé–Hoover thermostat (relaxation time = 1 ps) were used. The smooth particle mesh Ewald (PME) method (PME) was used to calculate short- (cutoff = 9 Å) and long-range Coulombic interactions using RESPA integrator. The conformations captured within the simulation trajectories were exported every 5 ps. The system stability was assessed through root mean square fluctuations (RMSF), hydrogen bond analysis, radius of gyration (Rg) and a histogram for torsional bonds after the completion of the MD simulation.

## 4. Conclusions

In summary, this study demonstrated the high potential of a new *Nocardiopsis* strain, CG3, to produce a wide array of structurally diverse secondary metabolites with antibacterial and antifungal activities. These compounds can be obtained through the biosynthetic pathway, such as kenalactams A–E (**39**, **43**, **69**, **75** and **78**), mitomycin C (**12** and **13**) and the *p*-terphenyl derivative (**65**), or via biotransformation of soybean aglycone isoflavones, such as glycitein and genistein, to 6,7-dimethoxy-3-(4-methoxyphenyl)chromen-4-one (**50**), 6,7-dimethoxy-3-phenylchromen-4-one (**54**) and 5,7-dimethoxy-3-(4-methoxyphenyl)chromen-4-one (**52**)**.** In addition, an in silico study indicated that the three isoflavone derivatives **50**, **52** and **54** are potent inhibitors of the aromatase enzyme (CYP19A1). Therefore, the preparation of fermented soybean milk rich in compounds **50**, **52** and **54** can be recommended for postmenopausal women in order to decrease breast cancer risk.

## Figures and Tables

**Figure 1 antibiotics-11-00657-f001:**
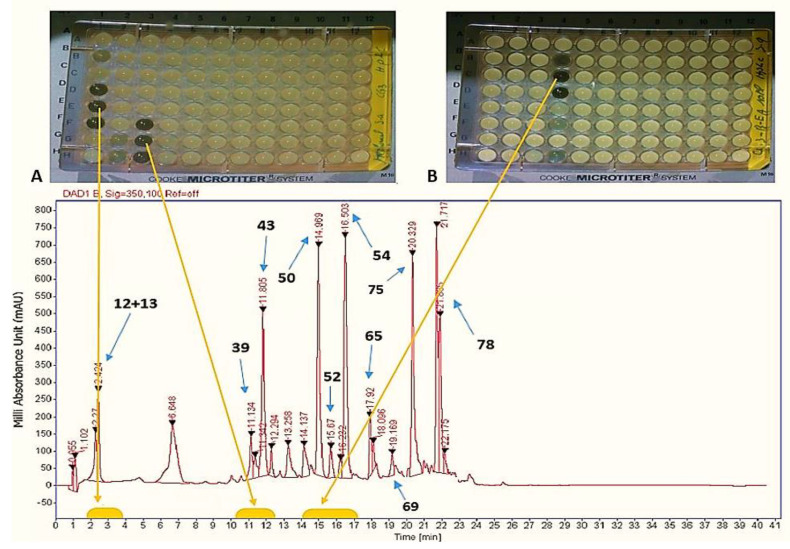
HPLC chromatogram of the crude extract from SM fractionated and tested against (**A**) *Staphylococcus aureus*; (**B**) *Mucor hiemalis*.

**Figure 2 antibiotics-11-00657-f002:**
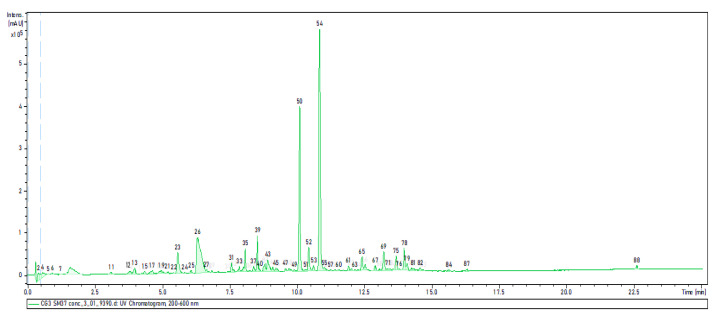
LC-UV-MS profile of crude extract prepared from the culture of strain CG3 in SM.

**Figure 3 antibiotics-11-00657-f003:**
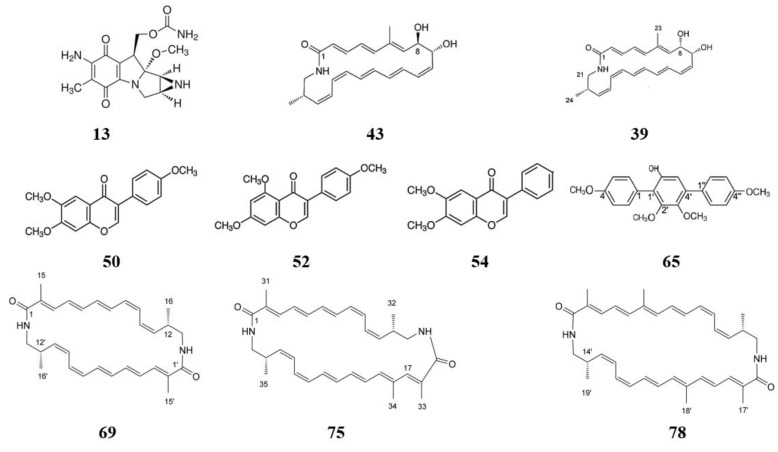
Structures of compounds identified from crude extract prepared from the culture of strain CG3 in SM.

**Figure 4 antibiotics-11-00657-f004:**
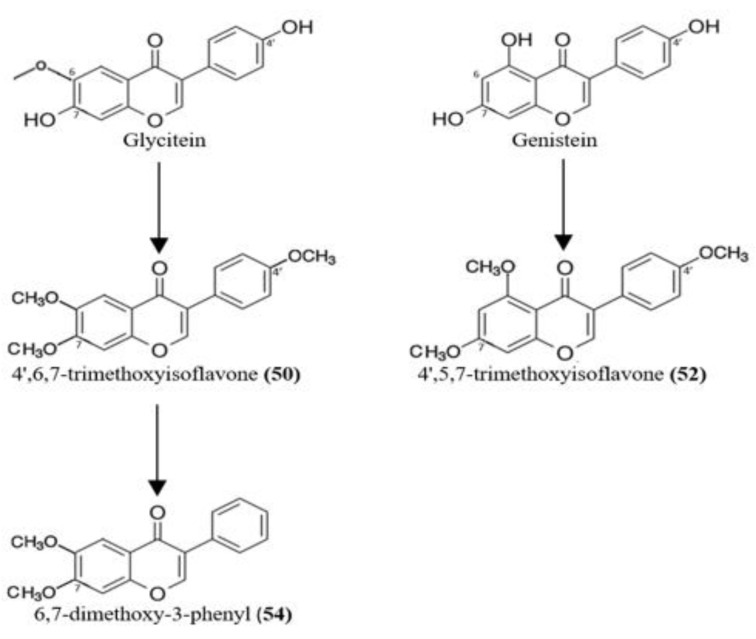
Proposed biotransformation of glycitiein to **50** and **54**, and genistein to **52**, respectively.

**Figure 5 antibiotics-11-00657-f005:**
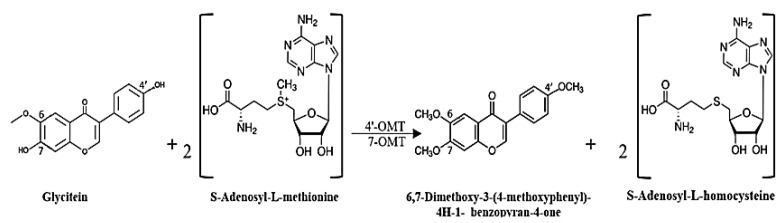
Possible biosynthetic pathway of 6,7-Dimethoxy-3-(4-methoxyphenyl)-4H-1benzopyran-4-one (**50**).

**Figure 6 antibiotics-11-00657-f006:**
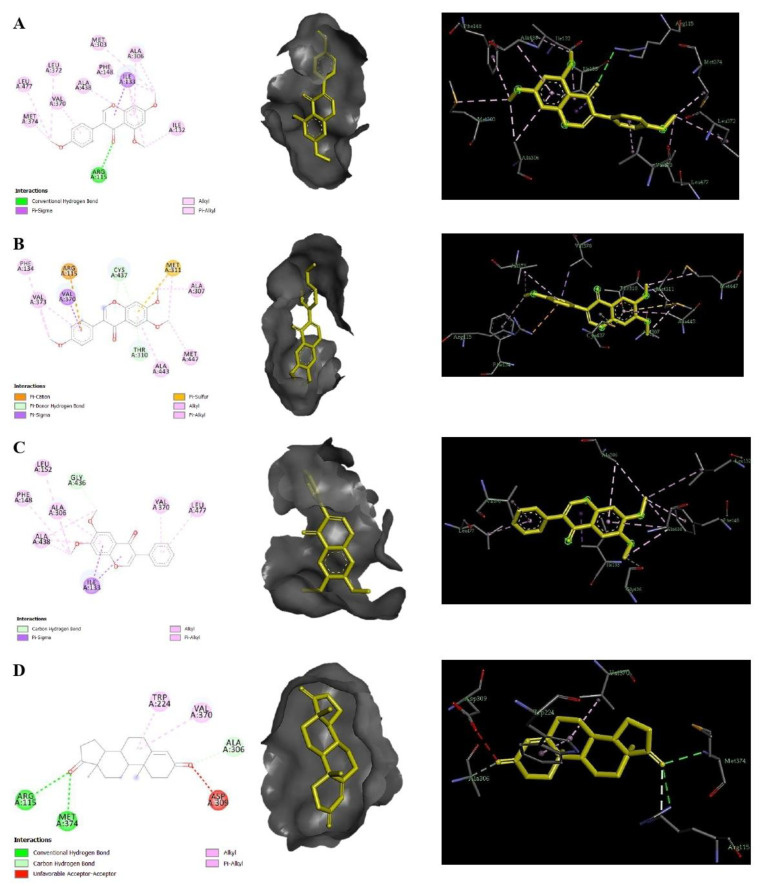
Interaction of **52** (**A**), **50** (**B**), **54** (**C**) and androstenedione (**D**) with the binding cleft of 3EQM, shown in 3D and 2D representations.

**Figure 7 antibiotics-11-00657-f007:**
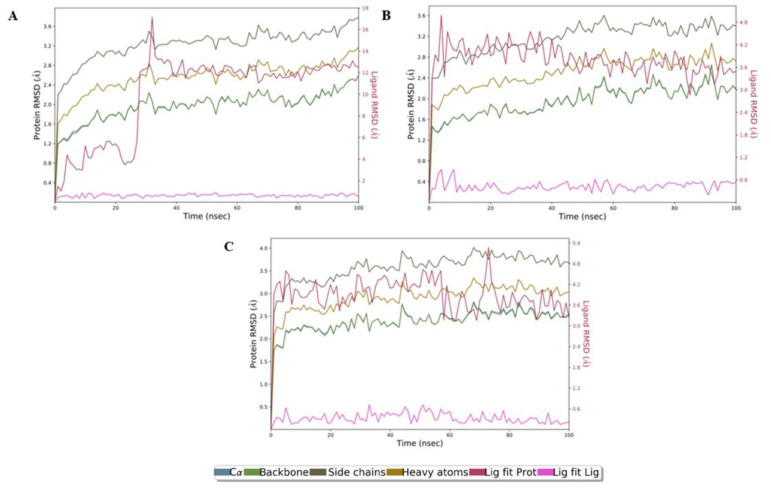
RMSD plot of human placental aromatase cytochrome P450 target and three compounds as a function of simulation time. (**A**) (50), (**B**) (52) and (**C**) (54). Color legends: Cα (blue color), side chains (green color), heavy atoms (yellow color), ligand with protein (dark pink color) and ligand with ligand (pink color).

**Figure 8 antibiotics-11-00657-f008:**
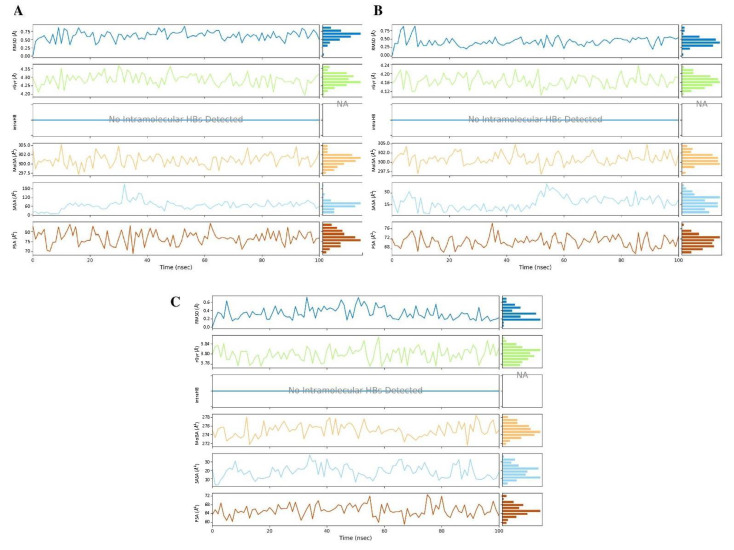
Various measures of the molecular dynamics simulations of three compounds, (**A**) (**50**), (**B**) (**52**) and (**C**) (**54**), within the pockets of human placental aromatase cytochrome P450 target.

**Figure 9 antibiotics-11-00657-f009:**
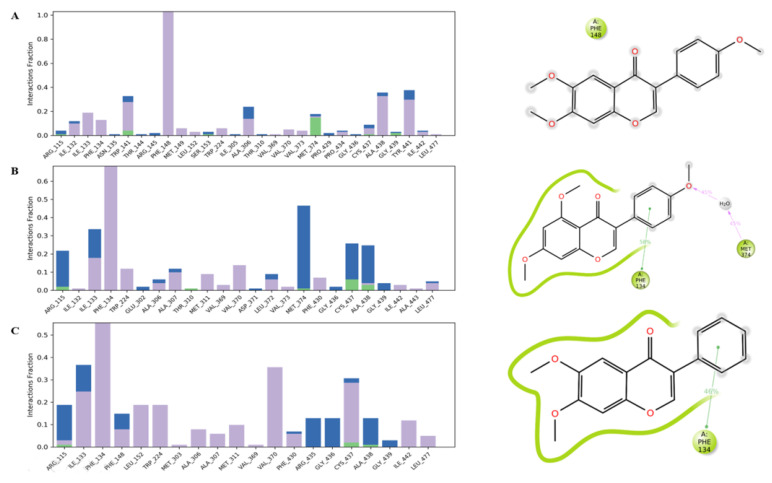
Intramolecular interactions of compounds **50** (**A**), **52** (**B**) and **54** (**C**) with human placental aromatase cytochrome P450 target. Color denotation: Hydrogen bonds (green), Hydrophobic contacts (purple), Water Bridge (blue).

**Table 1 antibiotics-11-00657-t001:** Antimicrobial activity of strain CG3 against human pathogenic microorganisms.

MIC (µg mL^−1^)										
	1	2	3	4	5	6	7	8	9	10
**SM**	2.08	66.66	-	1.04	0.52	2.08	0.52	66.66	66.66	16.66
**ISP2**	2.08	66.66	-	1.04	0.52	2.08	0.52	66.66	-	-
**NC**	-	-	-	-	-	-	-	-	-	-
**PC**	1.66	1.66	1.66	0.52	2.08	0.05	0.52	1.66	8.8	4.16

**1:***Escherichia coli*; **2:***Pseudomonas aeruginosa*; **3:***Klebsiella pneumoniae*; **4:***Chromobacterium violaceum*; **5:***Bacillus subtilis*; **6:***Staphylococcus aureus*; **7:***Micrococcus luteus*; **8:***Mycobacterium smegmatis*; **9:***Candida albicans*; **10:***Mucor hiemalis*; **NC:** negative control (methanol); **PC:** positive control (oxytetracycline).

**Table 2 antibiotics-11-00657-t002:** Binding energy values of three isoflavone compound derivatives, **50**, **52** and **54**, against aromatase (3EQM).

Compounds	Target Proteins (3EQM); Binding Energy (kcal mol^−1^).
5,7-Dimethoxy-3-(4-methoxyphenyl)chromen-4-one (**52**)	−7.5
6,7-Dimethoxy-3-(4-methoxyphenyl)chromen-4-one (**50**)	−7.1
6,7-Dimethoxy-3-phenylchromen-4-one (**54**)	−7.3
Androstenedione	−9.8

## Data Availability

Not applicable.

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
