# Peer review of "Metabolic Profile, Biotransformation, Docking Studies and Molecular Dynamics Simulations of Bioactive Compounds Secreted by CG3 Strain"

_antibiotics, 2022, doi:10.3390/antibiotics11050657_

Round 1

Reviewer 1 Report

Nocardiopsis is known  as one  of the pharmaceutically and biotechnologically important genera that attract natural products , mainly for its ability to produce a wide variety of secondary metabolites accounting for its wide range of biological activities.

In summary, several new derivatives are postulated but a full structural proof of the new structures is not provided (one of the new ones would be desirable) . It gives insight that there are many more derivatives out there, than formerly anticipated.
The biosynthesis discussion is plausible but highly speculative.

 At a closer look, the correlation of the compound and the activity was never established by the authors. 

Here are some remarks: 

1. Preparation of suspension: flasks...were inoculated with mature spores of strain CG3 ,  what is the amount ?

2. After two weeks incubation in a rotary shaker at 37°C and 160-180 rpm. Is it 160 or 180 rpm. 

3. Phylogeny of the producer strain, 
Since the 16S rRNA is – concerning the genus Nocardiopsis – not really sufficient to resolve the species, I would recommend – particularly if  the genome sequence is available – The so-called TYGS software to determine the species more reliably ! or alternatively show that the ANI-calculations consider all and solely type-strains.

4.In my opinion, for bioactivity assays, the choice of ESKAPE group (Enterococcus faecium, Staphylococcus aureus,  Klebsiella pneumoniae, Acinetobacter baumanii, Pseudomonas aeruginosa and Enterobacter spp.) witch  are the most  virulent germs, recently reported by the World Health Organization as able to form a community like  organization, designated as biofilm , would have been more targeted .

5. Maybe the reader should be first introduced to the bioactivity of compounds, 6,7-Dimethoxy-3-(4-methoxyphenyl)-4H-1- 29, benzopyran-4-one and 4H-1-Benzopyran-4-one, 6,7-dimethoxy-3-phenyl, as well as the  4',5,7-trimethoxyisoflavone    (possibly in the intro-section ?) , and that this activity is possibly in line with the observed bioactivity of the crude extract ?

6. Concerning Figure 8. Various measures of the molecular dynamics simulations of three compounds, A 397 (50), B (52) and C (54 ......, Are the   details available free of charge via the internet , letters are so small and the figure is not sufficiently clear.

7. Figure 6 and 7: same observation.

8. On page 12, using Starch Casein Agar (SCA) medium, after incubation at 30°C for 14 days [36]. Why did the strain take  14 days  to grow? by optimisation, we can reduce to, at least, 10 days of incubation.

Author Response

Response see saparate file.

Reviewer 2 Report

The manuscript has three parts of work, part 1. isolation compounds from the fermentation broth of secondary metabolites of Nocardiopsis CG3. part2. Proposed biotransformation of isoflavone, and part 3 docking studies of isoflavone.

Part 1 is substantially correlated with discovery of secondary metabolites. By the way, the most important compounds, Kenalactams A−E was published in JNP2019. Part 2 is useful for researchers of microbial chemists to refer when using soybean as recipe of media which will complicate the secondary metabolites analysis. Part 3 is no related with antibiotics, which can be published in other journal. I suggest to delete this part of work and submit to another journal.

Other errors found in the manuscript.

1, Line 82-84 Table1 should be rearranged, especially the strain name should be in the same line, by the way, the indicator strains name appeared for the first time, so, the full name of strain should be presented. You can use 1, 2, 3, but in the end of Table, full name of indicator strain name should be presented.

2, Line 98-99, fractionated via analytical HPLC and tested against S. aureus (Figure 1a) and M. hiemalis 98 (Figure 1b), subsequently analyzed by LC-HRESIMS.

But where is Figure 1a and 1b? in the figure 1, A and B presented as A : S. aureus ; B : M. hiemalis

3, Line 105, The result represented in Figure 1 indicated the presence of three active areas (1D, 105 1F), (3F, 3H), (4B, 4D), please delete “represented” and add “in 96-well microplates” in the end of the sentence.

4, the arrows should not point to peaks, pointing to yellow area is more reasonable.

5, Line 122, figure usually below the chart, which should be unified with Figure 1 and move to L123

6, Line 126, the peak number 12 (tR = 3.77 min) and 13 (tR= 3.95 min) in Figure 2. Peaks 12 and 13, but in Figure2, there are no Peak12, where it is?

7, Line162 (δH 7.01–8.24, δC 154.9, –101.2) why there is a comma between 154.9 and 101.2?

8, Line 165, HSQC-DEPT?

9, Line 167, NMR spectroscopid?

10, Line 157, soybean medium? You only used two media: SM medium, ISP2 medium, WHY?

11, Line 203, Compound 52 was structurally close to genistein, Figure5 should be mentioned in this line.

12, Line 240, Figure 5. Proposed biotransformation of glycitiein to 50 and 54, and genistein to 52, respectively. Sounds better.

13, why 238 inside of fig5

14, Line279,delete 2019

15,Fig 3 Structures of compounds 69, 75 and 78 should replaced by more clear structures

16, Table 2, the name “4H-1-Benzopyran-4-one, 6,7-dimethoxy-3-phenyl (54)” should revised according to IUPAC

17, left side of Figure 6 and Fig7 and 8 and 9 is indistinct

18, Line 452 soybean SM medium, please unified SM medium, soybean medium in Line72, Line157 and soybean SM medium in the whole paper

19, Line 469, “96-well plates” replace better by “96-well microplate”

20, Line 464, 20 µl should be twenty microliters

21. Line 480, Afterward, the plate was inoculated with previously tested pathogenic microorganisms. How to add bacteria and what is solvent or media, please state or write in detail.

Author Response

Response see separate file.

Reviewer 3 Report

The manuscript from Messaoudi and coworkers is of high interest on the searching for natural compounds with biologic potential to be studied and used for future applications. They performed a biological-guide fractionation of bacterial extract from CG3 strain as promising niche to search and isolate molecules with biological activity against several pathogenic microorganisms, and the findings are interesting. However, the manuscript needs to be improved and some comments are mentioned below.

  • The study does not imply metabolomics; I strongly recommend to authors changes the title by “metabolic profile … instead metabolomics profile” (Although authors, in materials and methods section mention this, I suggest changing it).

  • Line 26. Replace “High performance liquid chromatography coupled with high resolution electrospray ionization mass spectrometry (HRESIMS)” by High performance liquid chromatography coupled to high resolution mass spectrometry (HPLC-ESI-HRMS).

  • Line 80. MIC values were calculated by using bacterial competition or extract. Please, check it and correct if it is necessary.

  • Figure 1. It is not a spectrum; it is a chromatogram. What are the x and y axis on the chromatogram?

  • Photos resolution (figure 1) needs to be improved.

  • Line 112-114. It is not clear if the line refers to the third peak isolated between 14.50-16-50 min or the three peaks isolated from the SM medium. Please, check it.

  • Figure 2. Please, replace HRESIMS by LC-UV-MS profile.

  • Line 125-126: What was the criteria used by the authors to assign first zone of the chemical profile showed in figure 1 in relation to peaks 12 and 13, figure 2? It is not clear.

  • Line 127: What mean UV-VIS maxima? Do authors refer to UV spectrum?

  • Line 139: what is the bibliographic citation?

  • Line 140-141. Please, check the previous comment regarding the peaks assignment from figure 1 and figure 2. The same for lines 154-156.

  • Line 167. Correct the word “spectroscopid by spectroscopic”.

  • Line 217. Where are described these results?

  • Line 226: Replace P-terphenyl by p-terphenyl.

  • Line 267. Include the bibliographic citation.

  • Line 431. Replace “characterisation” by “characterization”

  • Experimental setup included bioassay-guided fractionation of the bacterial extract against several microorganisms of study. However, after chemical isolation of the compounds, why bioassays were not performed with the isolated compounds? (or synthetized). This will help to complement the experiments and results regarding active compounds from strain CG3.

  • Where is the conclusion of the study?

  • It is missing the supplementary material to review structure elucidation of compounds isolated, please, include this material.

Author Response

Response see separate file

Reviewer 4 Report

This manuscript describes the characterization of a number of bioactive compounds from Actinobacteria isolate CG3. Crude extract from this isolate was found to have antibacterial and anti fungal activity and specific compounds within the extracts were characterized for their activity against bacteria and fungi.

Overall, the manuscript is largely sound but there are some issues that should be addressed. 

  1. The genus-species names of the bacteria and fungi from the initial bioactivity screen should be given (they are not found until the methods section at the end of the manuscript). An explanation of why these species were chosen should be given (if there is one). Were these randomly chosen or do they represent a broad array of characteristics (gram-negative versus gram-positive, aerobe vs anaerobe, different sensitivity to other antimicrobials, etc). This will help the reader assess how broadly active this extract is.
  2. There are additional spots from the 96-well plates shown in Fig 1A and B that seem to show partial inhibition as well (in part A, 1B and 2G-H; in part B, 4F and 4H). Are these being investigated, perhaps against other organisms in your screen?
  3. Page 4, line 115 mentions a supplemental figure that was not provided. This figure could actually be part of the main manuscript. It is an important result that strongly implicates which peaks possess antifungal activity. Other supplementary figures/tables were also unavailable for review.
  4. It is unclear how the peaks in Fig 2 are correlated with the peaks in Fig 1, especially in those cases in which there appear to be multiple peaks in Fig 1 that would be within a single fraction (methods states fractions taken every 30 sec). How is it determined which peak corresponds to the bioactive compound?
  5. It would be helpful to the reader to include a table of MIC values for the various compounds tested.
  6. Testing of compound 65 against the various Fusarium species is out of the blue. There is no rationale provided for why this was done.
  7. Compound 65 was found to have both antifungal and antibacterial activity, yet the fractionation shown in Fig 1 A and B did not reveal any compounds that had activity against both S. aureus and M. hiemalis. If other testing of compounds was done, it would be useful to show this (perhaps in a table) to give the reader an overall view of which compounds were active against which species tested. 

Author Response

Response see separate file.
